:ᐧᐤ: PLOS | ONE

# Conspecific and interspecific stimuli reduce initial performance in an aversive learning task in honey bees (*Apis mellifera*)

**Christopher A. Varnon**[1]*, **Christopher W. Dinges**[2], **Adam J. Vest**[2], **Charles I. Abramson**[2]

**1** Laboratory of Comparative Psychology and Behavioral Ecology, Department of Psychology, Converse College, Spartanburg, South Carolina, United States of America, **2** Laboratory of Comparative Psychology and Behavioral Biology, Department of Psychology, Oklahoma State University, Stillwater, Oklahoma, United States of America

* Christopher.Varnon@Converse.edu

**Data Availability Statement:** All relevant data are within the manuscript and its Supporting Information files.

## Abstract

The purpose of this experiment was to investigate whether honey bees (*Apis mellifera*) are able to use social discriminative stimuli in a spatial aversive conditioning paradigm. We tested bees' ability to avoid shock in a shuttle box apparatus across multiple groups when either shock, or the absence of shock, was associated with a live hive mate, a dead hive mate, a live *Polistes exclamans* wasp or a dead wasp. Additionally, we used several control groups common to bee shuttle box research where shock was only associated with spatial cues, or where shock was associated with a blue or yellow color. While bees were able to learn the aversive task in a simple spatial discrimination, the presence of any other stimuli (color, another bee, or a wasp) reduced initial performance. While the color biases we discovered are in line with other experiments, the finding that the presence of another animal reduces performance is novel. Generally, it appears that the use of bees or wasps as stimuli initially causes an increase in overall activity that interferes with early performance in the spatial task. During the course of the experiment, the bees habituate to the insect stimuli (bee or wasp), and begin learning the aversive task. Additionally, we found that experimental subject bees did not discriminate between bees or wasps used as stimulus animals, nor did they discriminate between live or dead stimulus animals. This may occur, in part, due to the specialized nature of the worker honey bee. Results are discussed with implications for continual research on honey bees as models of aversive learning, as well as research on insect social learning in general.

## Introduction

In this paper, we investigate whether honey bees (*Apis mellifera*) are able to associate conspecific or interspecific cues with shock in an aversive conditioning paradigm. Honey bees have long been a popular species of study in fields such as behavioral ecology, comparative psychology, and neurophysiology. Colonies are often considered superorganism units [1–3] and thus a substantial body of literature investigates the rich social repertoire of honey bees. Despite possessing a relatively small nervous system of around one million neurons [4, 5], honey bees are able to direct hive mates to food sources [6, 7], use collaborative decision-making to select

**Funding:** This material was based upon work which was supported in part by National Science Foundation grants DBI 0552717, DBI 1263327, OISE 1545803, GRF 1144467 (received by CIA), and by National Institutes of Health grant P20GM103499 (received by CV). The funders had no role in study design, data collection and analysis, decision to publish, or preparation of the manuscript.

**Competing interests:** The authors have declared that no competing interests exist.

new hive locations [8], discriminate between related and unrelated individuals [9–12], and maintain a hygienic living area by removing dead or diseased individuals [13, 14]. These and other tasks are coordinated across a somewhat flexible, age-based, division of labor [15–17].

Another major area of research examines the learning ability of bees, focusing primarily on appetitive conditioning. In appetitive experiments, bees either learn to associate neutral stimuli with desirable events, or learn to alter their behavior to produce a pleasant outcome. The proboscis extension response conditioning procedure has long been used to study appetitive conditioning in controlled laboratory conditions [18–21], while other methods use free-flying, foraging bees when ecological validity is of greater importance [22, 23]. Research using these methods has produced a number of findings related to topics such as visual and olfactory sensation [24–26], perception of time [27], conditioned taste aversion [28], learning of abstract concepts [29, 30], quantity discrimination [31, 32], and the neurophysiology of memory [33–35].

Substantially less attention has been given to aversive conditioning, where bees learn that aversive stimuli are presented in association with other stimuli, or as a consequence for behavior [36, 37]. While reviews specifically comparing use of appetite and aversive techniques are not available, aversive literature is somewhat less common. A growing body of research investigates aversive conditioning in a shuttle box, where unrestrained bees learn to alter their behavior to reduce shock or other aversive stimuli in a small runway. Research using this method has explored areas such as learning differences between drones and workers [38], learned helplessness [39], visual discrimination [40], modulation of phototaxis [41], detection of narcotics [42], and the roles of the dopamine, octopamine and the mushroom body in aversive learning [41, 43, 44].

A relatively unexplored area of research is the intersection of social behavior and aversive learning. While a number of studies show that foraging and appetitive learning can be affected by social stimuli [21, 45–47], we are aware of only two studies that investigate aversive learning in relation to a social stimulus. The first investigation showed that low levels of exposure to isopentyl acetate, the principal component of alarm pheromone, improves learning in a shuttle box procedure while exposure to geraniol, the main component of the social homing pheromone, did not have any effect on learning [48]. The second study found that isopentyl acetate increased sensitivity to shock in a sting extension response paradigm, while geraniol decreased aversive responsiveness [49]. These two studies are the first to demonstrate that social stimuli can affect aversive learning, however, as they only investigates the effect of the primary components of pheromones, not the complex visual, tactile, and olfactory stimuli related to interactions with an intact animal, the study of social stimuli in aversive conditioning remains an open area.

The following experiment employs a group design with 14 groups to address the relative lack of aversive conditioning research when compared to appetitive procedures, as well as an almost complete absence of social aversive conditioning studies. In this experiment, we use a shuttle box method to investigate spatial aversive learning using intact animals as social stimuli. Subjects are thus provided with a complex range of social stimuli, similar to what they may encounter in natural interactions. Across several groups, bees learn that either shock or safety is associated with a hive mate. As comparisons, we also investigate whether bees are able to distinguish between a conspecific hive mate, or an interspecific paper wasp, as well as whether the bees respond differently if the stimulus animals are alive or deceased. Additional groups are included as standard and social controls.

## Materials and methods

### Subjects

Experimentally naïve worker honey bees (*Apis mellifera*) were collected from a single hive in Stillwater, Oklahoma (36.1156˚ N, -97.0584˚ W) for use as subjects in this experiment

(n = 268). Subjects were ultimately divided into 14 groups, with approximately 19 subjects per group. We used this sample and group size as this group size is extremely common in honey bee learning research, and as this experimental protocol is relatively novel, the literature was insufficient to permit a power analysis that would suggest a specific sample size. Additionally, our previous research [38, 39] with this general method suggested this sample size was adequate. Foraging bees were collected from a feeder containing 50% sucrose solution (weight/volume) approximately 50 meters away from the hive. All bees were collected from this single hive so that the experiment could test the ability of worker bees to use hive mates as a discriminative stimuli while avoiding issues related to antagonistic behaviors between bees from different hives. After collection, the bees were housed communally in a wire mesh carrier and were fed a 50% sucrose (weight/volume) solution. Bees were moved to a separate mesh carrier after acting as subject. Any bees that were complexly immobile once placed in the apparatus were discarded from the experiment. As Oklahoma State University does not require an ethics institutional review for non-threatened invertebrates, no specific review or permits were required to collect subjects for the present study.

## Social discriminative stimuli

Bees used as conspecific social stimuli, hereby referred to as *stimulus bees*, were collected and maintained in the same manner as subject bees. The stimulus bees were moved to a separate mesh carrier after acting as a stimulus and were not reused as subjects. The interspecific social stimuli were paper wasps (*Polistes exclamans*), hereby referred to as *stimulus wasps*, and were collected along with their nests from several locations on Oklahoma State University campus during fall. Colonies of stimulus wasps were collected from high traffic areas where they are frequently removed by staff. As colonies of North American paper wasps survive for only a single year, with the fall marking the end of a colony's life cycle, we expect collection had minimal impact on the campus wasp population. Each nest of wasps was maintained separately in a small plastic container, and the wasps' natural nests were glued to the top of the container. Wasps were provided with water, 50% sugar sucrose solution (w/v), and small portions of cricket and mealworm meat *ad libitum*. Stimulus wasps were returned to their nest after acting as a stimulus in an experimental session. As Oklahoma State University does not require an ethics institutional review for non-threatened invertebrates, no specific review or permits were required to collect stimulus animals for the present study.

## Apparatus

The apparatus consisted of two shuttle boxes and a control unit (see Fig 1). Two independent subjects were run simultaneously, one in each shuttle box, during each session. Subjects were run in pairs for the sake of efficient data collection, and were not able to interact to any degree. The general design of the shuttle boxes was modeled after those employed in [38, 39, 43]. The walls of the shuttle boxes were constructed from bars of HDPE plastic while the ends of the shuttle boxes were constructed from 3D printed ABS plastic. The ends served as both doors to permit subject entry and as stimulus chambers. The stimulus chambers were separated from the subjects' area in shuttle box with a metal mesh. The mesh permitted odor transfer and tactile interaction, including allowing antennae to protrude through the mesh, but did not permit subjects or stimulus animals to cross the barrier. The inner dimensions of the stimulus chambers were 50 x 50 x 20 mm (length x width x height). During initial testing of the apparatus we were able to observe that the stimulus animals could rotate in the chamber, but could not move closer to, or farther from, the subjects' area.

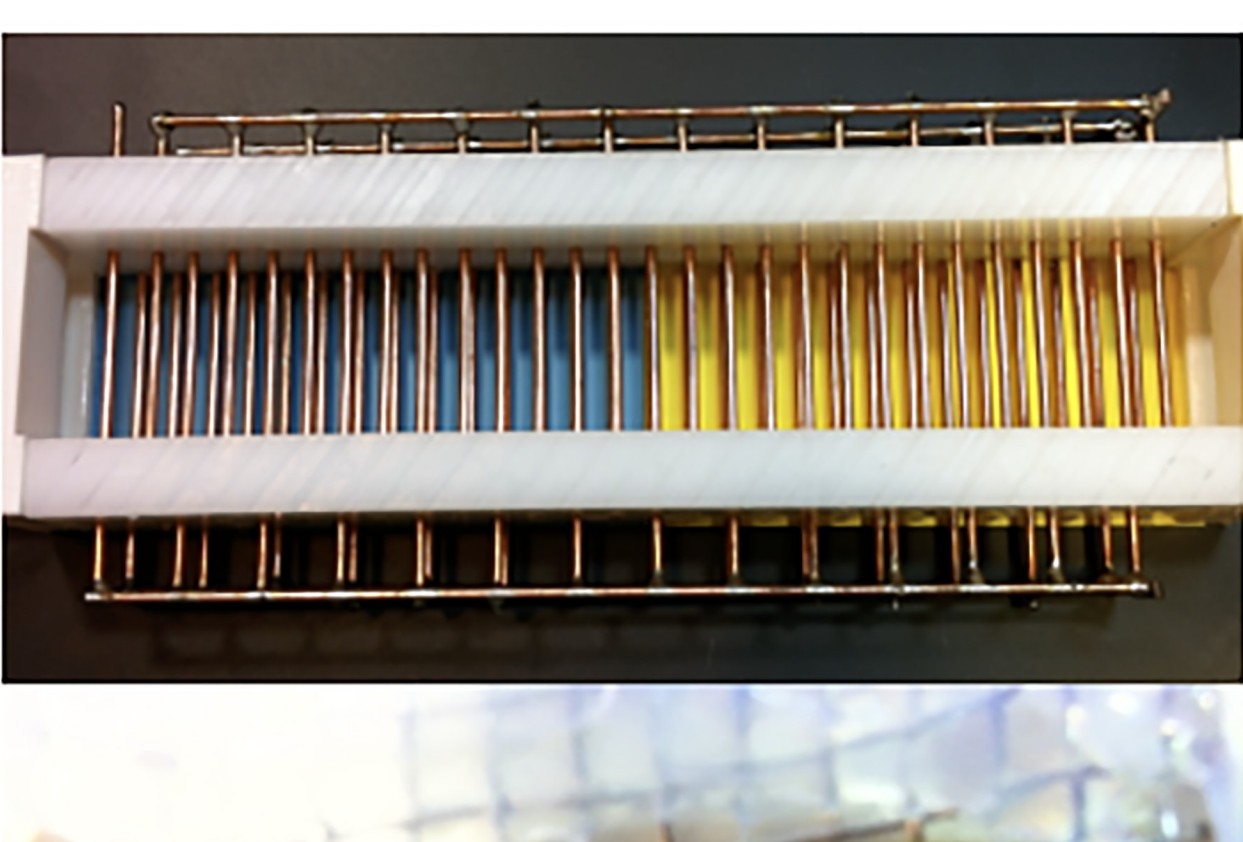

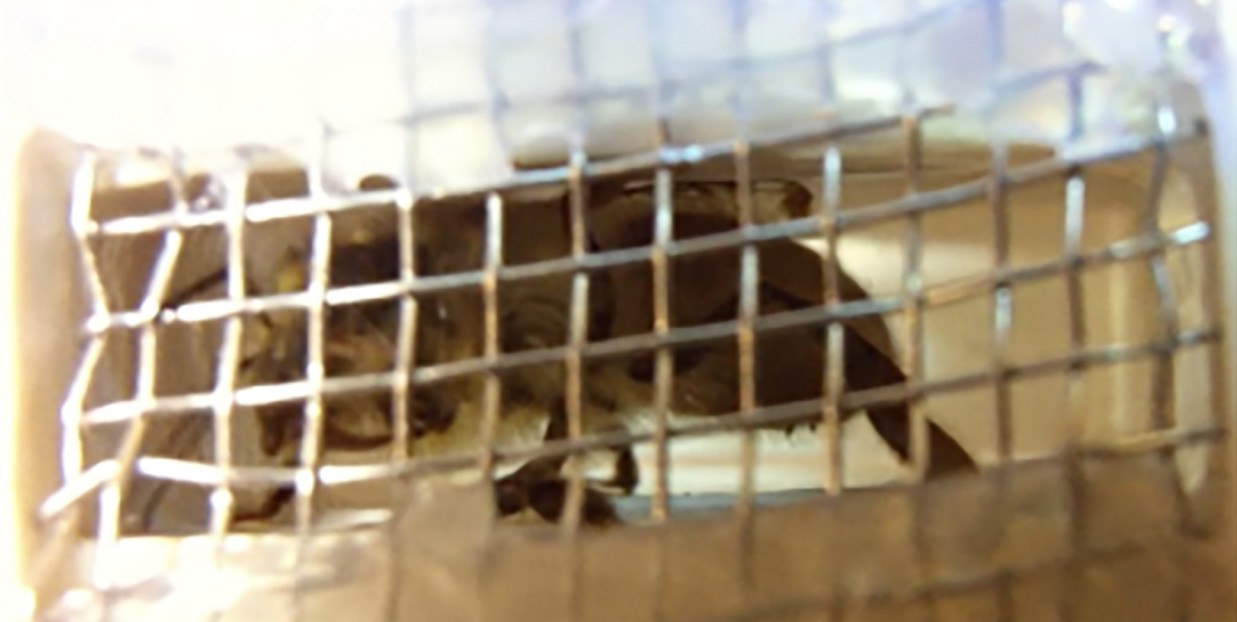

**Fig 1. Shuttle box apparatus.** The top image shows a single shuttle box, disconnected from the electronics unit for clarity. Here, the shuttle box is set up for a blue/yellow discrimination. The bottom image is taken from the subject's perspective inside the shuttle box and shows a stimulus bee in the stimulus chamber.

The ceiling and floor of the shuttle boxes consisted of a series of brass rods wired to form a shock grid. The rods were spaced 5.0 mm apart (center to center) with 3.5 mm gaps between the rods. Alternating rods were connected such that a bee could be shocked when it touched two adjacent rods. Clear acrylic strips were placed above the ceiling shock grid and below the floor shock grid to ensure that smaller subjects could not escape through the small gaps

between the rods. The inner dimensions of the shuttle boxes measured 145 x 50 x 20 mm (length x width x height), ensuring that bees could move with little restriction but were always in contact with either the ceiling or floor shock grid. A pair of 38 kHz modulated infrared beams in the center of each shuttle box was used to detect the location of the subjects. We used modulated infrared beams as they are resistant to ambient infrared interference from artificial lights and sunlight.

The control unit consisted of a Propeller Experiment Controller [50, 51], and a series of LEDs and switches to allow users to control the parameters of each session. During an experiment, the Propeller Experiment Controller read the experimental parameters from the user interface, generated the 38 kHz infrared beam signal, detected the subjects' locations, and provided 9 VDC shock to the entire shock grid through electromechanical relays based on the subjects' locations. This level of shock was based on a number of previous experiments [e.g., 38, 39] and ensured that subjects were not physically impaired and did not sting or vomit, which could cause an increase in conductivity and shock received. After each session was completed, the Propeller Experiment Controller created a data spreadsheet on a microSD card containing detailed information on the instances when subjects broke the infrared beams and the duration that subjects spent on each side of the shuttle boxes. The data spreadsheets were transferred to a computer at the end of each day after sessions were completed, though the computer was not required to conduct experimental sessions.

## Procedure

Each experimental session lasted nine minutes, and was divided into 18 30-second bins for sake of data analysis. Previous research indicates that this session length is adequate for observing learning, and changes in behavior are often apparent after a single minute [38]. During sessions, the shock grid activated if the subject was on the incorrect side of the shuttle box, as determined by group. Two subjects were run simultaneously in each shuttle box.

Subjects were randomly assigned to one of 14 groups. For all groups, the location of shock and stimuli were counterbalanced. See Table 1 for an overview of the group-design. The first set of groups were considered *standard controls*. These groups were used to replicate some existing findings of the honey bee aversive conditioning literature. In the *spatial* group, the shuttle boxes were placed on a plain grey background without stimulus animals present in the

**Table 1. Experimental design.**

| Group Category | Group | n |
|---|---|---|
| Standard Control | Spatial | 20 |
| Standard Control | Shock on blue | 19 |
| Standard Control | Shock on yellow | 19 |
| Social Control | Bee baseline | 17 |
| Social Control | Wasp baseline | 18 |
| Social Control | Bee and wasp baseline | 20 |
| Conspecific | Safe by live bee | 20 |
| Conspecific | Shock by live bee | 20 |
| Conspecific | Safe by dead bee | 18 |
| Conspecific | Shock by dead bee | 18 |
| Interspecific | Safe by live wasp | 20 |
| Interspecific | Shock by live wasp | 20 |
| Interspecific | Safe by dead wasp | 20 |
| Interspecific | Shock by dead wasp | 19 |

stimulus chambers. Shock was presented on either the front or the rear side of the shuttle box, with shock location counterbalanced across subjects. This group was used to assess spatial learning in the absence of other explicit cues. In the *shock on blue* and *shock on yellow* groups, the shuttle boxes were placed on a colored background where the color under the one half (either the front or the rear) of the shuttle box was blue, and the color under the other half of the shuttle box was yellow. Shock was presented on the blue side of the shuttle box for the shock on blue group, while shock was presented on the yellow side of the shuttle box for the shock on yellow group. Shock location was counterbalanced across subjects. We used these color discrimination groups to see whether bees could learn a non-social cue to avoid shock.

The second set of groups acted as *social controls* and investigated responses to hive mates or wasps in the absence of shock. In the *bee baseline* group, one live stimulus bee was placed in a stimulus chamber, but no shock was delivered. Similarly, the *wasp baseline* group used one live wasp in a stimulus chamber, and did not deliver shock. The *bee and wasp baseline* group used a bee in one stimulus chamber, a wasp in another stimulus chamber, and did not deliver shock.

The third set of groups investigated whether bees could use a *conspecific* as discriminative stimuli. These groups included a live or dead stimulus bee from the same hive as the subject. All procedures with conspecific stimuli were run with the shuttle boxes on the grey background to investigate the effects of the social cues without adding additional visual cues. In the *safe by live bee* group, subjects received shock when on the opposite side of the apparatus as the stimulus bee, while in the *shock by live bee* group, subjects received shock when they were on the same side as the stimulus bee. The *safe by dead bee* and *shock by dead bee* groups functioned similarly, except that the stimulus bee was deceased. All dead stimulus bees were killed by freezing for several hours and were thawed before the experimental session began. Killing bees in this manner is consistent with other experiments that study both social and necrophoresis responses [13, 52–54]. Neither live nor dead stimulus bees were used for more than one day.

The final set of groups investigated whether bees could use an *interspecific*, in this case the wasp *P. exclamans*, as discriminative stimuli. These groups included a live or dead stimulus wasp. As with the conspecific groups, shuttle boxes were placed on the grey background. The *safe by live wasp*, *shock by live wasp*, *safe by dead wasp*, and *shock by dead wasp* groups all mirrored their conspecific group counterparts, except that a stimulus wasp was used instead of a stimulus bee. All dead stimulus wasp were killed by freezing for several hours and were thawed before the experimental session began. Neither live nor dead stimulus wasps were used for more than one day.

## Measurement and analysis

**Measurement.**　We analyzed two measures. The first, *correct-compartment restriction (CCR)*, refers to the duration the subjects spent in the correct compartment of the shuttle box. Honey bees constantly move inside the shuttle box, and thus CCR refers to the time their movement is restricted to the correct compartment, with an emphasis that they still may be exploring and pacing in that compartment. We report CCR in terms of percent, thus 50% CCR is a chance level of response indicating that a bee's behavior is similar on both sides of the shuttle box. Our second measure, *activity*, concerned the overall activity of the subject, as measured by the number of times the infrared beams in the center of the apparatus were broken. Higher activity levels indicate more movement near the center of the apparatus, but not necessarily side-crossing.

**Regression analysis.**　We first analyzed CCR and activity using a repeated measures linear regression via generalized estimating equations [e.g., 28, 55–57]. All regression analysis were conducted through the StatsModels package [58] included in the Anaconda distribution of

Python [59], a free scientific analysis distribution of the Python programming language (python.org). To analyze CCR, we first subtracted 50 from all scores so that the chance level of CCR (50, or 50%) became 0. Positive CCR scores thus refer to an improvement above chance, while negative CCR scores refer to a decrease below chance. This adjustment allows regression parameters to easily refer to difference between observed CCR and expected chance level of CCR. As activity level does not have an intuitive chance level of response, we analyzed z scores of activity. The z scores were created with respect to the mean of activity scores for the entire experiment, including all groups and bins. This allows regression parameters for activity to refer to an increase or decrease compared to the mean of all activity scores.

For all regression analyses, as the subjects' behavior may change across time as a function of learning and we sought to investigate differences across groups, we included group, bin, and the group x bin interaction in all models. We used interceptless regressions that considered each level of the group variable to be a mutually exclusive categorical variable. The regression results thus compare the dependent variable to 0 (chance for CCR, or average for activity), as well as compare each groups' bin effect to 0 (no change across bin).

This interceptless regression method can be obtained by a number of manners including recoding the group variable and running the regression multiple times so that each group acts as the reference level once, by use of intercept removal commands in statistical software packages, or by manually recalculating regression parameters. Given the speed of modern computers we chose the first method of rerunning the regression. After obtaining initial regression parameters, our analysis next provided pairwise comparisons between each groups and between each group x bin interaction. In regressions with standard intercepts, pairwise comparisons between two parameters can be seen when the first parameter is included in the intercept. The estimate, standard error, confidence intervals and *p*-value of the second parameter represent the difference between the parameters. We simply saved these comparisons when rerunning the regression. Pairwise comparisons can also be calculated manually by finding a difference between parameter estimates, then creating a z score by dividing the difference between the estimates by the square root of the sum of the squared standard errors of the estimates [60, 61]. Regardless of method, the result is the same.

We found our interceptless method to be highly interpretable. For example, consider regression with a three-group categorical variable as the only parameter. The standard form of displaying the regression results would display: 1) a comparison to zero for the first group, 2) a comparison between the first and second groups, and 3) a comparison between the first and third groups. Our method displays a more complete set of results: 1) a comparison to zero for the first group, 2) a comparison to zero for the second group, 3) a comparison to zero for the third group, 4) a comparison between the first and second groups, 5) a comparison between the first and third groups, and 6) a comparison between the second and third groups.

Given the somewhat arbitrary nature of *p*-values and multiple comparison correction techniques, we did not include a single, specific manner to adjust the significance threshold when making multiple comparisons. Instead, we encourage the reader to select their own initial threshold and adjustment criteria. Then, the reader is free to interpret our analysis in the manner they believe is the most appropriate. One simple and common approach would be adjusting the traditional alpha value of 0.05 using a Bonferroni correction. This would involve dividing the alpha value by the number of pairwise comparisons being made. For example, in the bottom three rows of Table 2 we compare the learning rates (group x bin interactions) of the spatial, shock on blue, and shock on yellow groups. The provided *p*-values could either be compared to the traditional alpha value of 0.05, or a Bonferroni corrected threshold of 0.0167 (0.05 divided by three comparisons). A wide number of other of multiple comparison methods could also be used. Regardless of the interpretation of the *p*-values, other aspects of the analysis

**Table 2. Control groups CCR analysis.**

| Parameter | Estimate | Standard Error | 95% Confidence Intervals | | p-value |
|---|---|---|---|---|---|
| Spatial | 4.483 | 4.330 | -4.003 | 12.969 | 0.300 |
| Shock on blue | -1.975 | 3.972 | -9.761 | 5.810 | 0.619 |
| Shock on yellow | -9.162 | 3.343 | -15.714 | -2.609 | 0.006 |
| Spatial x Bin | 0.644 | 0.306 | 0.045 | 1.244 | 0.035 |
| Shock on blue x Bin | -0.403 | 0.300 | -0.992 | 0.186 | 0.180 |
| Shock on yellow x Bin | 0.592 | 0.282 | 0.040 | 1.144 | 0.036 |
| Pairwise Comparison | | | Difference | Standard Error | p-value |
| Spatial: Shock on blue | | | 6.458 | 5.876 | 0.272 |
| Spatial: Shock on yellow | | | 13.645 | 5.470 | 0.013 |
| Shock on blue: Shock on yellow | | | 7.186 | 5.192 | 0.166 |
| Spatial x Bin: Shock on blue x Bin | | | 1.048 | 0.429 | 0.015 |
| Spatial x Bin: Shock on yellow x Bin | | | 0.053 | 0.416 | 0.899 |
| Shock on blue x Bin: Shock on yellow x Bin | | | -0.995 | 0.412 | 0.016 |

remain consistent. For the sake of simplicity, we will primarily focus on the parameter estimates, but when considering p-values we will compare them to the traditional alpha level of 0.05.

**Ordinal analysis.** In addition to the regression analysis, we also analyzed CCR with an ordinal analysis technique following the overall method outlined by James Grice [62]. For detailed discussion and examples see [23, 38, 63–66]. This technique does not consider aggregate descriptive statistics or rely on comparisons to standardized distributions. Instead, all observations are compared directly, and comparisons between observed and randomized data are used in place of comparing a test statistic to a standard distribution. This approach provides an intuitive way to analyze data that is not subject to the many assumptions and interdisciplinary criticisms of traditional null-hypothesis significance testing [67–71].

In our ordinal analysis of CCR, we analyzed each group separately. For each subject in a group, we conducted a pairwise comparison of CCR for each bin. We used this method to test three models: 1) CCR increases across bin, with earlier bins having CCR less than or equal to subsequent bins; 2) CCR decreases across bin, with earlier bins having CCR greater than or equal to subsequent bins; and 3) CCR does not change across bin, with earlier bins having CCR equal to subsequent bins. For each model, a percent correct classified (PCC) value was obtained. The PCC value represents the percent of observations that fit a model prediction. The PCC value can thus be considered similar to measure of model fit.

After obtaining a PCC value, the data for a group was then randomized (the entire group's data set was shuffled across subject and bin), and the ordinal analysis was run again with the randomized data. This time a random PCC value was recorded. This process was repeated 1,000 times. The PCC value for the observed data and the randomized PCC values were compared to produce a chance value (c-value). The c-value represents the probability of the randomized PCC values being equal to or greater than the observed PCC value. In some ways, the c-value can be interpreted similarly to a p-value, in that a low c-value suggests a PCC value was very unlikely to occur by chance. However, unlike a p-value, no population assumptions are needed to interpret a c-value, nor is an arbitrary significance point given. Overall, interpretation of this method relies on 1) considering whether the model is valid, 2) considering the PCC value for the model, and 3) considering the c-value for the model. While this and other techniques are freely available in the Observation Oriented Modeling software [72], we conducted the analysis as described above in Python (S1 Ordinal analysis code).

## Results

### Standard control groups

Fig 2 shows correct-compartment restriction (CCR) and activity levels in the spatial, shock on blue, and shock on yellow groups. While bees in the spatial group quickly improved above the 50% chance level of CCR, bees in the color discrimination groups seldom performed above chance. Generally, it appears that the presence of color decreased the bees' ability to reduce shock. For the shock on blue group, CCR begins at nearly chance, but decreases somewhat across the experiment. For the shock on yellow group, however, the bees appear to have a preference for yellow, initially responding below chance. However, this bias is less apparent later in the experiment as CCR improves above the chance level, likely due to learning. In terms of activity level, bees in the spatial group were somewhat less active in the center of the shuttle box than bees in the color discrimination groups, but no other clear tendencies were observed.

S1, S2 and S3 Figs in the supporting material show heat map plots of CCR and activity levels of the spatial, shock on blue, and shock on yellow groups that illustrate the behavior of each individual bee. While variability across groups was high, a few interesting tendencies can be observed. Most notably is that several subjects performed exceptionally well in the spatial group, with frequent CCRs of nearly 100% often accompanied by very low activity near the center of the apparatus. In the color discrimination groups, low levels of CCR are also often accompanied by low activity levels. Generally, we observed low level of activity consisted of pacing within a small area, not immobility.

Table 2 shows the results of a regression analysis of CRR including the spatial, shock on blue, and shock on yellow groups, as well as interactions between these groups and bin. Generally, these analyses support earlier inspection of the graphs. The differences between the spatial, shock on blue, and shock on yellow groups' initial CCR can be seen both in the parameter estimate indicating difference from the 50% chance level of CCR, and the pairwise comparison

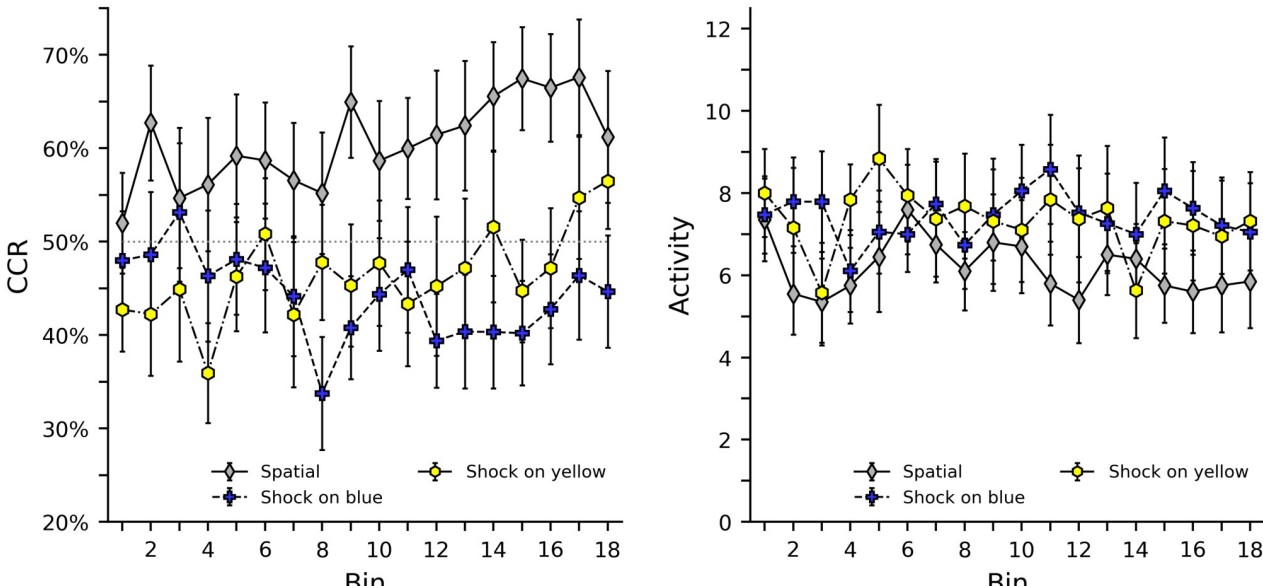

**Fig 2. Correct compartment restriction (CCR) and activity levels in the spatial, shock on blue, and shock on yellow groups.** The left plot shows average CCR values with error bars indicating standard error of the mean. Note the axis is truncated between 20% and 75% to provide a clearer view of the data. The dotted grey line in the center of the graph indicates the 50% chance level of response. The right plot shows average and standard error of activity levels, as defined as the total number of times a bee broke either infrared beam in the center of the shuttle box.

difference scores showing the disparity in CCR between the groups. The interactions of group and bin show the learning effects, and comparison between learning effects, for each group. As seen in Fig 2, initial CCR for the shock on yellow group is significantly below zero, and both the spatial and shock on yellow groups show significant learning effects. Table 3 shows a similar regression for z scores of activity levels, however, as seen in the graphs, few clear tendencies are observed. Note that for the pairwise comparisons shown in Tables 2 and 3, a multiple comparison correction to the significance threshold could be made. For example, a Bonferroni correction would suggest a threshold of 0.0167 (0.05 divided by three comparisons), for the pairwise comparisons of overall effects (three groups) and for the pairwise comparisons of learning effects (three group x bin interactions).

Table 4 shows an ordinal analysis of CCR across bins for all groups. For the spatial group, the increasing across bin prediction shows the highest percent correct classified (PCC) value and a very low $c$-value, indicating this prediction is the best fit for the data and that this level of PCC cannot easily occur by chance. The decreasing across bin prediction has a very high $c$-value, indicating that the PCC of 48.1 can very easily be obtained by random data. The equal prediction has a very low PCC value, but also a low $c$-value, indicating that CCR was rarely equal between two bins, but that this level of equality is very unlikely to be observed in randomized data. A careful inspection of the individual data (see S1 Fig) shows that once subjects reach 100% CCR, the CCR often remains at 100%. This may account for the low PCC value and $c$-value of the equal prediction. It is also important to note that the increasing across bin prediction includes cases where CCR is equal across two bins (i.e., the comparison is $\leq$ not $<$). Subtracting the equal PCC from the increase PCC, thus shows the percent of comparisons where a CCR from a bin is less than, but not equal to, a subsequent bin's CCR. For the spatial group, this is still higher than the decreasing prediction CCR. Similar results can be observed for the shock on yellow group. However, the shock on blue group appears to be a better fit for the decreasing prediction. Overall, the ordinal analysis supports the visual and regression analyses: the spatial group outperforms the other groups, and the shock on blue group performs the worst.

## Social control groups

Fig 3 shows CCR and activity levels in the bee baseline, wasp baseline, and bee and wasp baseline groups. For these groups, a side was considered "correct" if it contained a bee, did not

**Table 3. Control groups activity analysis.**

| Parameter | Estimate | Standard Error | 95% Confidence Intervals | | $p$-value |
|---|---|---|---|---|---|
| Spatial | -0.146 | 0.183 | -0.505 | 0.213 | 0.425 |
| Shock on blue | 0.014 | 0.188 | -0.354 | 0.382 | 0.940 |
| Shock on yellow | 0.078 | 0.188 | -0.290 | 0.447 | 0.676 |
| Spatial x Bin | -0.007 | 0.007 | -0.022 | 0.007 | 0.319 |
| Shock on blue x Bin | 0.002 | 0.008 | -0.013 | 0.017 | 0.783 |
| Shock on yellow x Bin | -0.006 | 0.008 | -0.021 | 0.009 | 0.403 |
| Pairwise Comparison | | | Difference | Standard Error | $p$-value |
| Spatial: Shock on blue | | | -0.160 | 0.262 | 0.541 |
| Spatial: Shock on yellow | | | -0.224 | 0.262 | 0.392 |
| Shock on blue: Shock on yellow | | | -0.064 | 0.266 | 0.809 |
| Spatial x Bin: Shock on blue x Bin | | | -0.010 | 0.011 | 0.372 |
| Spatial x Bin: Shock on yellow x Bin | | | -0.001 | 0.011 | 0.922 |
| Shock on blue x Bin: Shock on yellow x Bin | | | 0.008 | 0.011 | 0.432 |

**Table 4.  Ordinal analysis of change in CCR across bin.**

| Group | Increasing | | Decreasing | | Equal | |
|---|---|---|---|---|---|---|
| | PCC | $c$ | PCC | $c$ | PCC | $c$ |
| Spatial | 58.5 | 0.00 | 48.1 | 0.94 | 06.5 | 0.00 |
| Shock on blue | 51.8 | 0.29 | 55.2 | 0.01 | 06.9 | 0.00 |
| Shock on yellow | 57.7 | 0.00 | 49.8 | 0.73 | 07.5 | 0.00 |
| Bee baseline | 47.6 | 0.90 | 52.7 | 0.10 | 00.3 | 0.01 |
| Wasp baseline | 54.0 | 0.02 | 46.4 | 0.97 | 00.4 | 0.00 |
| Bee and wasp baseline | 50.9 | 0.36 | 50.1 | 0.50 | 01.0 | 0.00 |
| Safe by live bee | 57.4 | 0.00 | 45.1 | 0.99 | 02.5 | 0.00 |
| Shock by live bee | 53.1 | 0.05 | 49.5 | 0.66 | 02.7 | 0.00 |
| Safe by dead bee | 60.0 | 0.00 | 45.4 | 0.99 | 05.4 | 0.00 |
| Shock by dead bee | 59.3 | 0.00 | 47.2 | 0.97 | 06.5 | 0.00 |
| Safe by live wasp | 54.9 | 0.01 | 50.8 | 0.48 | 05.7 | 0.00 |
| Shock by live wasp | 55.5 | 0.00 | 47.5 | 0.93 | 03.0 | 0.00 |
| Safe by dead wasp | 59.4 | 0.00 | 45.5 | 0.99 | 04.9 | 0.00 |
| Shock by dead wasp | 55.3 | 0.00 | 47.0 | 0.95 | 02.2 | 0.00 |

contain a wasp, or both. Thus, CCR can be used to show preferences for these stimuli. Generally, it appears that the subject bees slightly preferred being near stimulus bees and away from stimulus wasps. This is somewhat observed in all three groups, but it is interesting that the bee and wasp baseline group does not show an additive effect, instead it shows the smallest effect. Subjects in the bee baseline group showed the highest activity level, while subjects in the wasp baseline group showed the lowest activity level. In this case, the activity level of the bee and wasp baseline is an intermediate of the other groups. Activity levels appear to slightly decrease across session for all groups. S4, S5 and S6 Figs in the supplemental material show heat maps illustrating CCR and activity of each individual bee.

Table 5 shows the results of a regression analysis of CCR including the bee baseline, wasp baseline, and bee and wasp baseline groups, as well as interactions between these groups and bin. Generally, these analyses support earlier inspection of the graphs; only weak, non-significant effects are observed, and the bee and wasp baseline group clearly does not show an additive effect of the bee baseline and wasp baseline groups. Table 6 shows a similar regression for z scores of activity levels. Subjects appear to be more active when a stimulus bee is present, but this effect decreases slightly over time when a wasp is also present. As with previous analysis a multiple comparison correction could be applied to the significance threshold for the pairwise comparisons in Tables 5 and 6.

An ordinal analysis of CCR for the social control groups can be seen in Table 4. The results here generally support the findings of visual and regression analysis. Any minor preferences observed in the bee baseline decrease across bin, while preferences in the wasp baseline increase across bin. It appears that neither an increasing nor decreasing pattern is a good fit for the bee and wasp baseline group.

Overall, it appears that subject bees may have a very minor preference for being near another bee. A slight preference for being away from the stimulus wasp is also present, and this is the only preference that seems to increase over time. A post-hoc single sample $t$ test of total CCR (bins summed; one measurement per subject) with all groups pooled, also found a small and nonsignificant tendency for the subject to be near the stimulus bee and away from the stimulus wasp: $t(54) = 1.804$, p = 0.0768, d = 0.243. Activity was also increased in the presence of another bee but decreases over time when a wasp is present. As all these effects appear

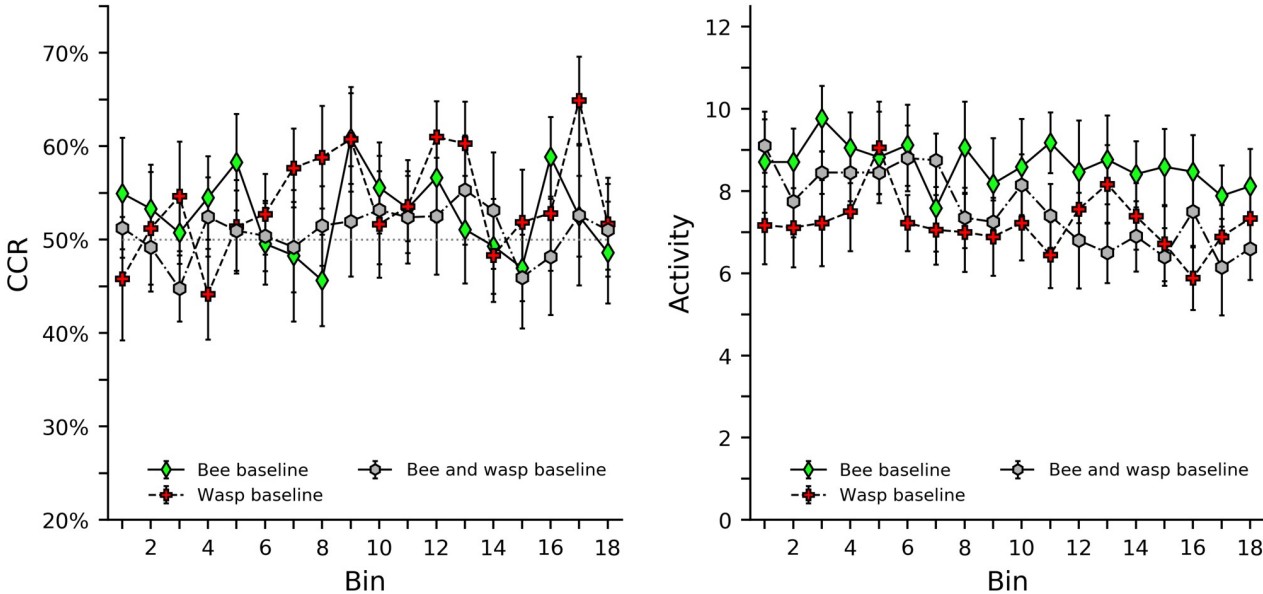

**Fig 3. Correct compartment restriction (CCR) and activity levels in the bee baseline, wasp baseline and bee and wasp baseline groups.** The left plot shows CCR values, with the side near the stimulus bee being considered correct, and the side near the stimulus wasp being considered incorrect. Error bars indicate standard error of the mean. Note the axis is truncated between 20% and 75% to provide a clearer view of the data. The dotted grey line in the center of the graph indicates the 50% chance level of response. The right plot shows average and standard error of activity levels, as defined as the total number of times a bee broke either infrared beam in the center of the shuttle box.

to be very minor, they do not hinder investigations of using stimulus bees or stimulus wasps as cues, as would strong, consistent biases.

## Conspecific and interspecific groups

Fig 4 shows CCR and activity levels in the safe by live bee, shock by live bee, safe by dead bee, and shock by dead bee groups. The spatial group was also included as a comparison. Generally, subjects in the conspecific groups initially perform below chance, then gradually improve

**Table 5. Social control groups CCR analysis.**

| Parameter | Estimate | Standard Error | 95% Confidence Intervals | | p-value |
|---|---|---|---|---|---|
| Bee | 3.770 | 2.661 | -1.446 | 8.986 | 0.157 |
| Wasp | 0.315 | 3.692 | -6.921 | 7.552 | 0.932 |
| Bee and wasp | -0.043 | 3.392 | -6.691 | 6.604 | 0.990 |
| Bee x Bin | -0.110 | 0.231 | -0.564 | 0.343 | 0.633 |
| Wasp x Bin | 0.396 | 0.282 | -0.158 | 0.949 | 0.161 |
| Bee and Wasp x Bin | 0.098 | 0.336 | -0.561 | 0.756 | 0.772 |
| Pairwise Comparison | | | Difference | Standard Error | p-value |
| Bee: Wasp | | | 3.455 | 4.551 | 0.448 |
| Bee: Bee and Wasp | | | 3.813 | 4.311 | 0.376 |
| Wasp: Bee and Wasp | | | 0.358 | 5.014 | 0.943 |
| Bee x Bin: Wasp x Bin | | | -0.506 | 0.365 | 0.166 |
| Bee x Bin: Bee and wasp x Bin | | | -0.208 | 0.408 | 0.610 |
| Wasp x Bin: Bee and wasp x Bin | | | 0.298 | 0.439 | 0.497 |

Bee baseline, wasp baseline and bee and wasp baseline groups are abbreviated as bee, wasp, and bee and wasp, respectively.

**Table 6. Social control groups activity analysis.**

| Parameter | Estimate | Standard Error | 95% Confidence Intervals | | p-value |
|---|---|---|---|---|---|
| Bee | 0.374 | 0.137 | 0.105 | 0.642 | 0.006 |
| Wasp | 0.060 | 0.173 | -0.279 | 0.398 | 0.731 |
| Bee and wasp | 0.351 | 0.089 | 0.176 | 0.526 | 0.000 |
| Bee x Bin | -0.010 | 0.007 | -0.023 | 0.004 | 0.155 |
| Wasp x Bin | -0.007 | 0.010 | -0.027 | 0.012 | 0.475 |
| Bee and Wasp x Bin | -0.030 | 0.009 | -0.046 | -0.013 | 0.001 |
| Pairwise Comparison | | | Difference | Standard Error | p-value |
| Bee: Wasp | | | 0.314 | 0.221 | 0.154 |
| Bee: Bee and Wasp | | | 0.023 | 0.164 | 0.890 |
| Wasp: Bee and Wasp | | | -0.292 | 0.195 | 0.134 |
| Bee x Bin: Wasp x Bin | | | -0.002 | 0.012 | 0.836 |
| Bee x Bin: Bee and wasp x Bin | | | 0.020 | 0.011 | 0.066 |
| Wasp x Bin: Bee and wasp x Bin | | | 0.022 | 0.013 | 0.086 |

Bee baseline, wasp baseline and bee and wasp baseline groups are abbreviated as bee, wasp, and bee and wasp, respectively.

across the session. Surprisingly, even though some minor preference for being near stimulus bees was observed in the bee baseline group, subjects do not appear to perform better when a stimulus bee predicts safety than when a stimulus bee predicts shock. It also does not appear that subjects make a distinction between live or dead stimulus bees. Instead, it appears that presence of any stimulus bee initially reduces performance, especially when compared to the spatial group. Activity levels also do not show a clear distinction between groups, but do show a slight overall decrease throughout the session, similar to what was observed for the social

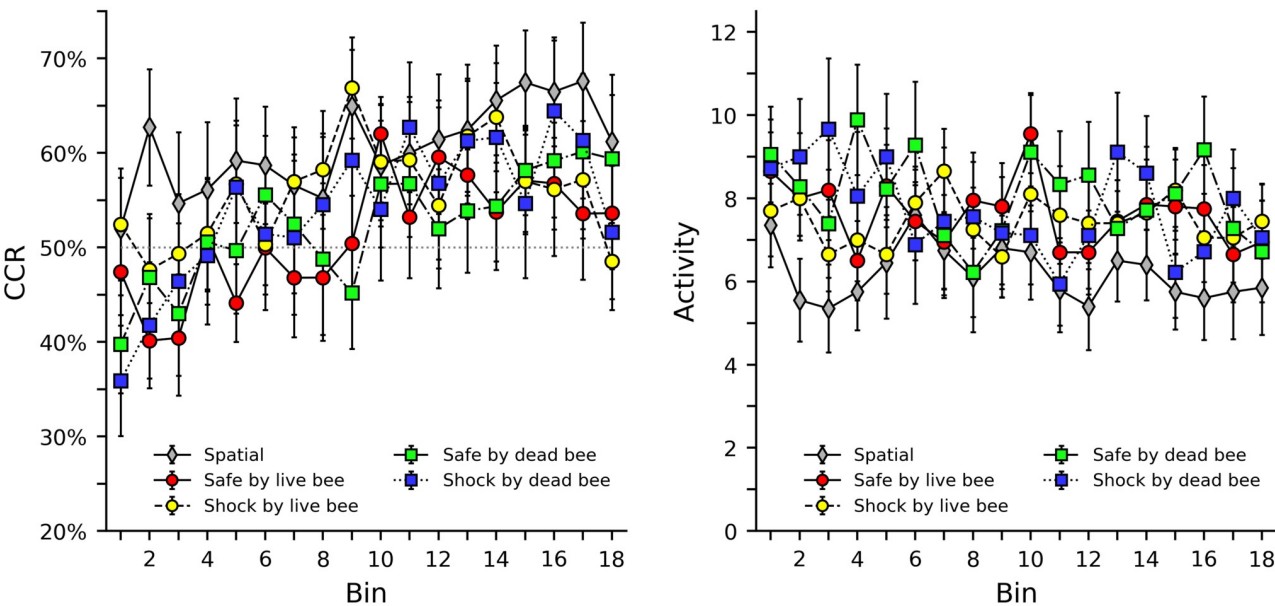

**Fig 4. Correct compartment restriction (CCR) and activity levels in groups that used another bee as a stimulus.** The left plot shows average CCR values with error bars indicating standard error of the mean. Note the axis is truncated between 20% and 75% to provide a clearer view of the data. The dotted grey line in the center of the graph indicates the 50% chance level of response. The right plot shows average and standard error of activity levels, as defined as the total number of times a bee broke either infrared beam in the center of the shuttle box.

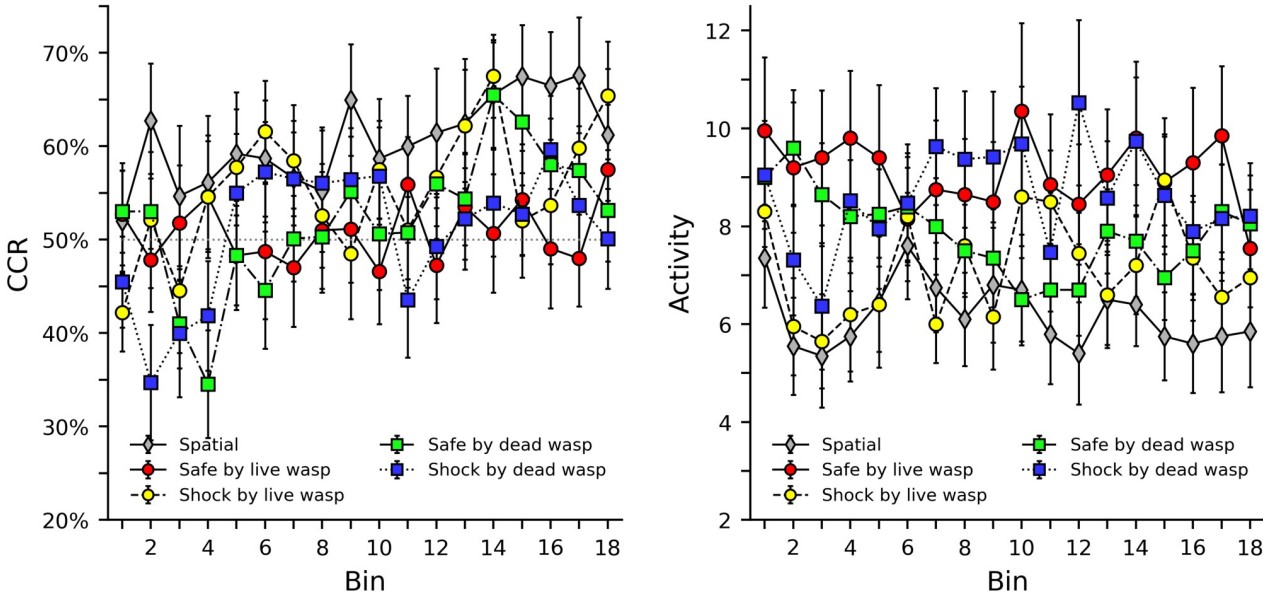

**Fig 5. Correct compartment restriction (CCR) and activity levels in groups that used a wasp as a stimulus.** The left plot shows average CCR values with error bars indicating standard error of the mean. Note the axis is truncated between 20% and 75% to provide a clearer view of the data. The dotted grey line in the center of the graph indicates the 50% chance level of response. The right plot shows average and standard error of activity levels, as defined as the total number of times a bee broke either infrared beam in the center of the shuttle box.

control groups (see Fig 3). S7, S8, S9 and S10 Figs in the supplemental material show heat maps of CCR and activity levels for the conspecific groups that illustrate the behavior of each individual bee. Generally, the same tendencies are observed as in Fig 4, but it can also be seen that extreme CCR levels are occasionally accompanied by low activity levels.

Fig 5 shows CCR and activity levels in the safe by live wasp, shock by live wasp, safe by dead wasp, and shock by dead wasp groups. The spatial group was also included as a comparison. Although the data appears more variable, similar tendencies are observed as with the conspecific groups. Subjects in the interspecific groups show initially low CCR levels, followed by gradual improvement, and this tendency is not distinct across group. Activity levels for the interspecific groups appear both higher and more variable than those of the conspecific groups. S11, S12, S13 and S14 Figs in the supplemental material show heat maps of CCR and activity levels for the interspecific groups that illustrate the behavior of each individual bee. As with the color discrimination and conspecific groups, high CCR levels are occasionally accompanied by low activity levels.

Tables 7 and 8 show the results of a regression analyses of CCR and activity levels, respectively. These analyses include all conspecific and interspecific groups, but also include the spatial group for comparison. Unlike previous tables, pairwise comparisons were not included. As the list of pairwise comparisons is rather lengthy, they are instead included in the supporting material as S5, S6, S7 and S8 Tables. For CCR, while the bees in the spatial group initially respond above chance, bees in the conspecific and interspecific groups initially respond below chance with the exception of the shock by live bee group. While this is fairly consistent, with only the shock by live bee group presenting an exception, these effects are not large or significant. The learning effects for all groups, however, were consistently positive, and also significant with the exception of the three weakest effects (shock by live bee x bin, safe by live wasp x bin, and shock by dead wasp x bin). It is worth noting that the regression does not show any

**Table 7. Experimental groups CRR analysis.**

| Parameter | Estimate | Standard Error | 95% Confidence Intervals | | p-value |
|---|---|---|---|---|---|
| Spatial | 4.483 | 4.330 | -4.003 | 12.969 | 0.300 |
| Safe by live bee | -6.678 | 3.543 | -13.622 | 0.265 | 0.059 |
| Shock by live bee | 2.501 | 3.794 | -4.935 | 9.937 | 0.510 |
| Safe by dead bee | -6.366 | 4.472 | -15.130 | 2.398 | 0.155 |
| Shock by dead bee | -5.862 | 5.527 | -16.694 | 4.970 | 0.289 |
| Safe by live wasp | -0.206 | 4.545 | -9.113 | 8.702 | 0.964 |
| Shock by live wasp | -1.298 | 4.726 | -10.561 | 7.965 | 0.784 |
| Safe by dead wasp | -6.172 | 3.663 | -13.352 | 1.007 | 0.092 |
| Shock by dead wasp | -5.244 | 4.284 | -13.641 | 3.154 | 0.221 |
| Spatial x Bin | 0.644 | 0.306 | 0.045 | 1.244 | 0.035 |
| Safe by live bee x Bin | 0.846 | 0.321 | 0.218 | 1.475 | 0.008 |
| Shock by live bee x Bin | 0.364 | 0.367 | -0.355 | 1.082 | 0.321 |
| Safe by dead bee x Bin | 0.918 | 0.375 | 0.182 | 1.654 | 0.014 |
| Shock by dead bee x Bin | 1.052 | 0.431 | 0.207 | 1.897 | 0.015 |
| Safe by live wasp x Bin | 0.114 | 0.309 | -0.491 | 0.719 | 0.711 |
| Shock by live wasp x Bin | 0.710 | 0.340 | 0.045 | 1.376 | 0.037 |
| Safe by dead wasp x Bin | 0.874 | 0.237 | 0.409 | 1.339 | 0.000 |
| Shock by dead wasp x Bin | 0.641 | 0.418 | -0.178 | 1.459 | 0.125 |

interesting similarities between groups. For example, the parameter estimates for groups where the stimulus animal predicts safety do not differ much from those that predict shock, nor do groups with live stimulus animals differ from those with dead stimulus animals, nor do groups with stimulus bees from those with stimulus wasps. The same general finding is true for the activity levels; most groups are similar and no overall differences between safe/shock, live/dead, and bee/wasp, are observed. The pairwise comparisons of CCR and activity level

**Table 8. Experimental groups activity analysis.**

| Parameter | Estimate | Standard Error | 95% Confidence Intervals | | p-value |
|---|---|---|---|---|---|
| Spatial | -0.238 | 0.168 | -0.568 | 0.092 | 0.157 |
| Safe by live bee | 0.080 | 0.163 | -0.239 | 0.399 | 0.625 |
| Shock by live bee | -0.053 | 0.141 | -0.329 | 0.224 | 0.708 |
| Safe by dead bee | 0.180 | 0.232 | -0.275 | 0.635 | 0.439 |
| Shock by dead bee | 0.201 | 0.276 | -0.339 | 0.741 | 0.466 |
| Safe by live wasp | 0.349 | 0.266 | -0.171 | 0.870 | 0.189 |
| Shock by live wasp | -0.208 | 0.170 | -0.540 | 0.125 | 0.221 |
| Safe by dead wasp | 0.195 | 0.179 | -0.157 | 0.546 | 0.278 |
| Shock by dead wasp | 0.115 | 0.228 | -0.332 | 0.562 | 0.614 |
| Spatial x Bin | -0.007 | 0.009 | -0.025 | 0.010 | 0.414 |
| Safe by live bee x Bin | -0.010 | 0.012 | -0.034 | 0.014 | 0.414 |
| Shock by live bee x Bin | 0.000 | 0.010 | -0.018 | 0.019 | 0.970 |
| Safe by dead bee x Bin | -0.011 | 0.016 | -0.043 | 0.020 | 0.479 |
| Shock by dead bee x Bin | -0.020 | 0.019 | -0.057 | 0.016 | 0.278 |
| Safe by live wasp x Bin | -0.007 | 0.016 | -0.037 | 0.024 | 0.674 |
| Shock by live wasp x Bin | 0.010 | 0.012 | -0.013 | 0.033 | 0.402 |
| Safe by dead wasp x Bin | -0.017 | 0.014 | -0.045 | 0.010 | 0.209 |
| Shock by dead wasp x Bin | 0.007 | 0.015 | -0.023 | 0.038 | 0.632 |

also show similar tendencies, and no substantial thematic difference between the various conspecific and interspecific groups emerge. The ordinal analysis in Table 4 support what the visual and regression analysis suggest about CCR. All conspecific and interspecific groups show an increase in CCR across bin in a manner that appears similar to the spatial group.

Taken together, it is clear that subjects in the conspecific and interspecific groups show similar improvement in CCR. However, although they are learning, the presence of conspecific or interspecific cues do not seem to aid subjects. Instead, any cue in addition to the spatial task initially reduces correct responding. Surprisingly, the bees in this experiment do not appear capable of using additional information to improve performance, and this is consistent across conspecific and interspecific groups.

## Discussion

### Overview

Generally, our results indicate that while bees in the spatial group display a rapid improvement in performance, the addition of any other stimuli (color, bee, or wasp) initially reduces performance. Bees in the spatial group showed a final level of performance of around 61% to 66% CCR (estimates derived the averaged data and from the analysis in Table 2, respectively). While only a few studies use comparable automated measurements to study aversive learning in honey bees, similar performance is consistently reported. Across four other papers, many including multiple experiments and groups, we found that bees in groups that responded above chance had an average CCR of around 61% [38–40, 44]. It appears that bees in our spatial group showed a typical response.

Although bees in the shock on blue and shock on yellow groups did not perform as well as bees in the spatial group, this is not surprising given the prevalence of color biases in bee research. Such biases are common in appetitive and foraging experiments, and have been observed in honey bees [73, 74], stingless bees [75], and bumble bees [76]. In an earlier shuttle box experiment using a previous version of our apparatus and a similar protocol, Dinges et al. [38] found that bees performed better when shock was associated with blue than when shock was associated with yellow. Similarly, Kirkerud, Schlegel, and Galizia [40] found that bees easily learned that safety was associated with green, but had trouble learning that green was associated with shock. Black et al. [77] explained this by demonstrating that biases acquired during natural foraging can persist in aversive conditioning procedures. Given that bees in our shock on blue and shock on yellow groups performed worse than bees in the spatial group, it is likely the bees had a history of feeding on both blue and yellow flowers. As bees in the shock on blue group responded worse throughout the experiment while bees shock on yellow group improved, our bees also likely had more experience feeding on blue than on yellow flowers.

While bees in the standard control groups responded similarly to findings reported in other literature, the low performance of the bees in the conspecific and interspecific groups may seem surprising. Unfortunately, there is no literature suitable for direct comparison. Generally, bees in conspecific and interspecific groups initially responded below chance, then slowly increased in performance throughout the experiment. This effect occurred consistently without respect to the predictive role, living status, or species of the stimulus animal. Our findings suggest that bees have difficulty using other bees or wasps as cues in this spatial aversive conditioning task and perform better overall when these stimuli are not available. This leads to two questions. First, why did the presence of another animal reduce initial performance compared to the spatial group and how was this overcome? Second, why were no differences observed in response to different types of animal stimuli?

## Reduced performance in the conspecific and interspecific groups

The reduced initial performance of the conspecific and interspecific groups compared to the spatial group suggests that the bees can detect, and are affected by, the stimuli animals. A major possibility for this difference is that the shuttling behavior of bees may initially sensitize, then habituate to the presence of other insects in the apparatus. Rapid shuttling behavior is common in this type of procedure [e.g., 40, 44], and the relationship between shuttling and reduced learning has been observed in other shuttle box experiments. For example, Dinges et al. [39] found that the shuttling behavior of bees prohibited use of traditional negative reinforcement (escape/avoidance) methods to study learned helplessness, and instead required a positive punishment method similar to the one used in the present experiment (i.e., shock is delivered contingent on entering a side). In the current experiment, the presence of other animals may cause increase shuttling as the bee investigates the situation, searches for better access to a hive mate, or attempts to an escape an apparatus containing wasp. This increased shuttling may be somewhat incompatible with learned performance, as has been observed in previous studies, and could cause a nearly chance level of correct response. Additionally, although we used a mild shock, the act of being shocked may physically inhibit a bee's ability to shuttle back to the correct side, resulting in more time spent on the incorrect side and thus a lower than chance level of correct response. Over time, the impact of the animal stimuli may decrease through habituation, leading to less overall movement, and facilitating behavior that is more directed by the aversive continencies.

Some evidence for this interpretation can be seen in the present experiment. The spatial group showed the highest average CCR and the lowest average activity of all groups. Conversely, all 11 groups with stimulus animals, even those without shock, showed much higher activity scores, and these scores decreased across the experiment with only three exceptions (shock by live bee–no change in activity; shock by live wasp and safe by life wasp–increase in activity). For groups with stimulus animals and shock, CCR scores were initially low and increased across the experiment, likely as the shuttling response habituated to the stimulus animal. Though we observed similar tendencies in earlier work using this method [39], the only other research using social stimuli in a spatial aversive conditioning paradigm found a different effect from our study. In a series of experiments, Avalos et al. [48] showed that exposure to components of alarm pheromone, but not homing pheromone, improves aversive spatial learning. It is possible that the general, holistic stimuli used in our experiment promotes increased activity that may be incompatible with the shuttle box task, while the isolated alarm pheromone used by Avalos et al. [48] causes a different response that enhances performance.

## Failure to discriminate between types of stimulus animals

We did not observe any fundamental distinctions in responses of subjects that received bees as stimuli compared to those that received wasps as stimuli, nor were clear differences observed when the stimulus animals were alive or dead. Both of these findings may be accounted, in part, by the specialized nature of the worker honey bee. It is possible that the bees we collected are not well suited to produce distinct behavior in response to nestmates compared to wasps. While bees and wasps often have antagonistic interactions and bees guarding the hive are known to make such distinctions [78, 79], the guard bee is a specialized role of worker bee that is unlikely to be present in our sample of bees that were collected at feeders. Forager workers may instead ignore the presence of other non-predator insects while collecting nectar and pollen. Alternatively, the presence of any pollinator insect (including other bees and paper wasps) may facilitate foraging at a particular location, as has been shown with bumble bees [52–54].

It is also possible that our sample of bees was also not well suited to discriminate between live and dead individuals. This discrimination may be reserved for undertaker bees, another specialized role that worker bees may take before becoming foragers [14, 80]. Social foraging research with bumble bees shows that dead conspecifics are indeed adequate cues to facilitate foraging [52–54]. As the traditional method of collecting worker bees captures bees while foraging, this sampling approach may also prevent undertaker bees from being collected.

## Future directions and conclusions

Our work contributes to a growing body of research on aversive learning in honey bees. Additionally, we present one of the only experiments investigating the relationship between social stimuli and aversive learning. The findings of our experiment suggest that, as with previous research using this method [39], careful consideration needs to be given to not only the learned performance of bees, but also to their tendency to rapidly pace inside a shuttle box. Our findings contribute to a body of research showing that while bees will learn the task, they are also affected by colors that may be related to their foraging history, and we provide a new findings suggesting that their tendency to pace may be enhanced in the presence of other insects, to the extent that it reduces initial performance.

The fact the bees in our experiment did not discriminate between live and dead stimulus animals also suggests that experiments investigating topics such as observational learning or social facilitation in insects should carefully consider what stimuli their subjects are actually responding to, and may benefit from using deceased stimulus animals as a control. Indeed, research on social facilitation of foraging in bumble bees has already shown that foragers prefer to select flowers occupied by another bee, regardless of whether the other bee is alive or dead [52–54]. Our research suggests similar effects may be observed in honey bees, and research on social learning in other taxa, such as *Drosophila*, may also benefit from considering this possibility.

Additional research will be required to determine specifically which social stimuli affect learning, and what manner these stimuli affect behavior. For example, could an isolated substance, such as the fatty acids found in comb wax that aids in hive mate recognition [10] evoke the reduced performance effect observed in this experiment? Combining such social substances with live, dead, or even dummy bees could help identify the cause of the effects we observed.

Future work should also consider using refined position and activity measurements such as the grid of infrared beams used by Kirkerud et al. [40] and Nouvian and Galizia [81], or a computer vision approach similar to that used by Marchal et al. [41] and Kimura et al. [82] to precisely measure the pacing behavior observed in shuttle box experiments. Additionally, investigations on the habituation to social stimuli discussed in this paper could also consider the many principles of habituation and sensitization outlined by Thompson and Spencer [83], Groves and Thompson [84], and Rankin et al. [85]. For example, the stimulus generalization principle suggests that habituation may to one stimulus may generalize to similar stimuli. In the case of social stimuli such as those used in the present experiment, habituation to one species of *Polistes* paper wasp may also cause habituation to other species of *Polistes*. These three papers [83–85] discuss a number of similar, relatively simple, but important principles that could be considered.

Finally, as honey bees are specialized not only by caste, but also by the current role of the worker bee, future research should consider explorations of multiple castes and worker roles. A growing body of research is already outlining learning differences between worker and drone castes [38, 48], but similar work is also needed to compare worker specializations and

ages. For example, bees that have matured into foragers but have not yet foraged may not display the color biases that are commonly observed, while newly emerged bees may show a completely different manner of responding than our subjects. We hope that our findings and suggestions will help facilitate continual research on honey bees as models of aversive learning.

## Supporting information

**S1 Table. Experimental groups CRR analysis—Group effects.**
(DOCX)

**S2 Table. Experimental groups CRR analysis—Bin interactions.**
(DOCX)

**S3 Table. Experimental groups activity analysis—Group effects.**
(DOCX)

**S4 Table. Experimental activity CRR analysis—Bin interactions.**
(DOCX)

**S5 Table. Experimental groups CRR analysis—Group effects.**
(DOCX)

**S6 Table. Experimental groups CRR analysis—Bin interactions.**
(DOCX)

**S7 Table. Experimental groups activity analysis—Group effects.**
(DOCX)

**S8 Table. Experimental activity CRR analysis—Bin interactions.**
(DOCX)

**S1 Fig. Heat map of CCR and activity levels of the spatial group.**
(DOCX)

**S2 Fig. Heat map of CCR and activity levels of the shock on blue group.**
(DOCX)

**S3 Fig. Heat map of CCR and activity levels of the shock on yellow group.**
(DOCX)

**S4 Fig. Heat map of CCR and activity levels of the bee baseline group.**
(DOCX)

**S5 Fig. Heat map of CCR and activity levels of the wasp baseline group.**
(DOCX)

**S6 Fig. Heat map of CCR and activity levels of the bee and wasp baseline group.**
(DOCX)

**S7 Fig. Heat map of CCR and activity levels of the safe by live bee group.** Note that one extreme activity score was truncated to 25 for sake of consistency across graphs.
(DOCX)

**S8 Fig. Heat map of CCR and activity levels of the shock by live bee group.** Note that one extreme activity score was truncated to 25 for sake of consistency across graphs.
(DOCX)

**S9 Fig. Heat map of CCR and activity levels of the safe by dead bee group.**
(DOCX)

**S10 Fig. Heat map of CCR and activity levels of the shock by dead bee group.**
(DOCX)

**S11 Fig. Heat map of CCR and activity levels of the safe by live wasp group.** Note that five extreme activity scores were truncated to 25 for sake of consistency across graphs.
(DOCX)

**S12 Fig. Heat map of CCR and activity levels of the shock by live wasp group.**
(DOCX)

**S13 Fig. Heat map of CCR and activity levels of the safe by dead wasp group.**
(DOCX)

**S14 Fig. Heat map of CCR and activity levels of the shock by dead wasp group.** Note that one extreme activity score was truncated to 25 for sake of consistency across graphs.
(DOCX)

**S1 Data.**
(CSV)

**S1 Ordinal analysis code.**
(PY)

## Acknowledgments

We are grateful for the useful comments of Paul Marchal and one anonymous reviewer. We would also like to thank Noelle Vallely and Charlie Beheler for their assistance with reviewing literature.

## Author Contributions

**Conceptualization:** Christopher A. Varnon, Christopher W. Dinges, Adam J. Vest, Charles I. Abramson.

**Formal analysis:** Christopher A. Varnon.

**Funding acquisition:** Charles I. Abramson.

**Investigation:** Christopher A. Varnon, Christopher W. Dinges, Adam J. Vest.

**Methodology:** Christopher A. Varnon, Christopher W. Dinges.

**Project administration:** Charles I. Abramson.

**Resources:** Christopher A. Varnon, Christopher W. Dinges.

**Software:** Christopher A. Varnon.

**Supervision:** Charles I. Abramson.

**Writing – original draft:** Christopher A. Varnon, Charles I. Abramson.

**Writing – review & editing:** Christopher A. Varnon, Christopher W. Dinges, Charles I. Abramson.

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
