## [Decision Letter · Decision Letter 0]

28 Oct 2019

PONE-D-19-27434

Conspecific and intraspecific stimuli reduce aversive learning in honey bees (Apis mellifera)

PLOS ONE

Dear Dr. Varnon,

Thank you for submitting your manuscript to PLOS ONE. After careful consideration, we feel that it has merit but does not fully meet PLOS ONE’s publication criteria as it currently stands. Therefore, we invite you to submit a revised version of the manuscript that addresses the points raised during the review process.

In particular, please address the statistical concerns of the reviewers, especially the concerns of reviewer 1 on sample size. 

We would appreciate receiving your revised manuscript by Dec 12 2019 11:59PM. To enhance the reproducibility of your results, we recommend that if applicable you deposit your laboratory protocols in protocols.io, where a protocol can be assigned its own identifier (DOI) such that it can be cited independently in the future. For instructions see: http://journals.plos.org/plosone/s/submission-guidelines#loc-laboratory-protocols

We look forward to receiving your revised manuscript.

Kind regards,

James C. Nieh, Ph.D.

Academic Editor

PLOS ONE

Journal Requirements:

1. 

2.  In your Methods section, please provide additional location information of the collections site, including geographic coordinates for the data set if available.

3.  In your Methods section, please provide additional information regarding the permits you obtained for the work. Please ensure you have included the full name of the authority that approved the collection site access and, if no permits were required, a brief statement explaining why.

Reviewers' comments:

Reviewer's Responses to Questions

**Comments to the Author**

1. Is the manuscript technically sound, and do the data support the conclusions?

Reviewer #1: Partly

Reviewer #2: Yes

2. Has the statistical analysis been performed appropriately and rigorously? 

Reviewer #1: Yes

Reviewer #2: Yes

3. Have the authors made all data underlying the findings in their manuscript fully available?

Reviewer #1: Yes

Reviewer #2: Yes

4. Is the manuscript presented in an intelligible fashion and written in standard English?

Reviewer #1: Yes

Reviewer #2: Yes

5. Review Comments to the Author

Reviewer #1: General points and major concerns:

This is a generally well-conducted, and thoroughly analyzed study that addresses a gap in the literature on honey bee aversive conditioning -- in particular, the ability of honey bees to use members of their own species or another species as a cue to facilitate learning. Given that honey bees are eusocial, it is reasonable to predict that individual bees should be attentive to the presence of a conspecific, and should therefore be capable of using this type of stimulus as a cue; however, what the authors show is somewhat more complex. They claim that, although the stimulus organism was noticed, the presence of another organism seems to have decreased performance, especially at the beginning of a trial, when subjects often performed below chance level. This is puzzling, given that what constituted "correct" was different across conditions -- so would have to mean that sometimes, the subject got closer to the stimulus animal, sometimes, further away (sometimes one side of the box, sometimes the opposite side), but that they seem to have been drawn toward making an incorrect choice in all cases.

The biggest limitation of the present study is the large number of groups (14) and time bins (18) employed, which would necessitate a larger sample size than the one the authors used -- particularly if the expected effect size is small. I am also concerned that the authors seem to have failed to control for the multiple comparisons they conducted. Since any effects observed appear to be rather small, it is understandable that the authors would not have wished to further diminish the probability of obtaining a significant effect; however, what comes out of the various analyses is often difficult to interpret, seeming to fluctuate randomly. Personally, I would need further assurances from the authors that any patterns seen in the present study are truly indistinguishable from noise.

Specific points:

Page 4, line 9, I think you meant "single" instead of signal.

Page 5, line 123, I am not sure I undestand. Do you mean that there are two bees simultaneoulsy in the same chamber? If so, would you not expect that to interact with whatever effect the conspecifics and interspecific animals had?

Another question about the shuttle box: you mention that the subjects were able to physically interact with the stimulus organisms on the other side of the mesh, but did you observe them doing this? How close were the stimuli, especially the dead stimuli, to the mesh? Is it possible

Page 23, line 439, I think a word is missing in the header.

You explain you small and lack of effects in the context of biases established during foraging. What would you predict would happen if you used newly-emerged bees instead of experienced foragers?

Throughout: I would avoid using the term "trend" unless you are referring to an actual trend analysis.

Reviewer #2: General comments:

In the manuscript entitled “Conspecific and intraspecific stimuli reduce aversive learning in honey bees (Apis mellifera)” (PONE-D-19-27434), the authors describe a series of experiments assessing if of honey bees can use conspecific (a bee from the same hive) or interspecific (a wasp) as cues for a place preference assay in a shuttle box. For these experiments the authors measured two variables: the “correct compartment restriction (CCR), indicating the preference of the bees, at a given time, for one or the other side of the shuttle box, and the “activity”, which indicates the number of time a bee passes from one side to the other of the shuttle box. In particular, the authors performed several experimental and control groups to test several possible effects of conspecific and interspecific cues on an aversive place preference learning task. The data are presented in a clear graphs and supplementary material figures present a graphical representation of the raw data. Along the paper, the authors proceed to a systematic description of results that makes it easy to understand the many analyses they performed. Addionally, the analysis of each experiment in three steps (visual description, regression statistics, ordinal statistics) is appreciated and allows the authors to not be constrained to only discuss the results in terms of statistical difference and p-values. Finally, the statistics used in this study are extensively explained in the material and methods section, which allows for a good understanding of the analyses.

However, some general and more specific aspects of the manuscript could be improved. On a general level, the bibliography cited in the manuscript is in several occasions rather dated or incomplete (see specific comments). The graphs could be improved with colors and error bars (the error bars could represent a confidence interval already computed in the “bootstrapping” ordinal analysis) for better visualization of the experimental data. Also, and most importantly, the authors seem to confuse the preference for a side or another (CCR) at a given time, and the evolution of the CCR value across time (change that can be a proxy of learning the conditioning rules of the experiment). This confusion is present at several places in the manuscript and should be remedied (see specific comments).

On the whole, despite some corrections needed to improve or correct the manuscript, this study has enough strength to be accepted for publication.

Specific comments:

- Throughout the manuscript, the interspecific group is sometimes mistakenly labelled intraspecific (title, introduction, etc…)

- Line 26, authors could mention that the experiment consists in a spatial conditioning, as other studies of the same kind in honey bees usually use visual or olfactive conditioning instead.

- In this study authors conclude that learning is similar in conspecific, interspecific and spatial groups (see Table S2; Spatial X bin : Other groups X Bin). However, in the title and abstract, they state several times that social cues reduced learning. In fact, social cue only generally reduced CCR, but doesn’t impair its improvement (learning). This should absolutely be remedied.

- Line 51, citations for nestmate and non-nestmate recognition and discrimination studies in the honey bee seem a bit dated. Maybe some other studies were published more recently and could be added.

- Line 58, the most recent review on the PER conditioning protocol is lacking from the citations and should be added (Matsumoto et al., 2012, J Neuro Methods)

- Line 61, a review article on visual conditioning in harnessed honey bees has been published recently and could be cited to illustrate the point (Avarguès-Weber and Mota 2016, J Physiol Paris)

- Line 62, recent publication by Howard and colleagues (science) have been published on numerical abilities in honey bees that would illustrate perfectly the point on quantity discrimination.

- Line 62, the most recent and up to date citations for the neurophysiology of memory in honey bees are (Müller 2012, Apidologie; Eisenhardt 2014, Learn & Mem).

- Line 65, to illustrate a broader spectrum of aversive conditioning laboratory studies, two citations could be added (Kirkerud et al., 2017; Marchal et al., 2019, Learn & Mem).

- Line 70, the list of the themes tackled using an aversive shuttle box kind of experimental setup could be widened citing Marchal et al., 2019 with the study of modulation of phototaxis.

- Line 71, Marchal et al., 2019 also qualify as a reference on the role of dopamine, octopamine and mushroom bodies in aversive learning.

- Line 72-81, the authors elaborate on the lack of studies on the modulation of aversive learning by social cues. This section could be improved by adding a statement concerning the studies that exist on shock responsiveness modulation by pheromones in honey bees, which might be interesting to discuss in the framework of this manuscript (Rossi et al., 2018, JEB).

- Line 123, the authors describe the setup as being able to perform the conditioning of two independent animals in parallel. How was this capacity used? Were animals of the same groups performed in parallel? Or on the opposite, were animals of different groups performed in parallel?

- The Apparatus section of the Material and Methods could benefit from a visual representation of the apparatus (that seem to be slightly different from the one used in previously published studies) to help understanding.

- Line 148, typo: Controlled should be replaced by Controller.

- Line 169-170, the description of the apparatus using the terms top half and bottom half is misleading as the terms top and bottom were previously used to describe what could be called floor and ceiling of the apparatus. This highlights again the improvement that a visual on the apparatus could ease understanding.

- Line 170, the text seems to suggest that a side of the apparatus was always associated to yellow and the other side to blue, indicating that the sides were not balanced with respect to the color stimuli. If this is the case, it might imply a spatial bias in the experiment.

- Line 227, contrasting with the rest of the statistical analysis description, the part explaining the z-score of activity might be missing clarity. It is not clear with respect to what mean value the z-scores are computed. Is it to the mean of all groups of a same experiment at a given bin? Is it to the mean of all bins of a single group?

- Line 296, the authors state that “the bias appears to dissipate” because the CCR is improving across bins. However, the increase is probably due to learning, while the bias is still present.

- Line 314, the low CCR bees in the color discrimination groups show low activity too, which means that the bees stayed in the shocked compartment (keeping receiving shocks), which might indicate that those bees were not feeling the shock, or behaviorally impaired. This possibility could be discussed.

- Line 367, “effect” could be replaced by “activity”.

- Line 368, “illustrate” should be replaced by “illustrating”.

- Line 382, the study does not seem to specifically test a general preference (all bins pulled) towards stimulus bee or stimulus wasp which could be interesting to assess (as a post hoc analysis for example).

- Line 435, “High” should be added before “CCR”.

- Line 448, as the conspecific and interspecific groups are compared to the spatial group, it might improve understanding of the data to make the spatial group appear on all conspecific and interspecific graphs for visual comparison.

- Line 488, what does the expression “initially reduces learning” mean? Learning being a change in behavior across time, the term initial learning does not seem to fit. This could be replaced by “reduces the overall CCR level”.

- Line 489, the authors state that the spatial group reached a final level of 66%. However, to be precise, this value is extracted from the model that fitted the data, not the measured behavior of the animals. The final CCR level that the spatial group reached is slightly lower and should be the one reported here.

- Line 498, stating that the study has been performed with the same experimental set up and protocol would strengthen the argument.

- line 503, the authors argue that the low CCR (compared to the spatial group) of bees in the shock on blue and shock on yellow groups is likely due to prior experience of bees on colored flower. However, if both groups were performed in parallel, and given the fact that they were exposed to blue and yellow cues at the same time in both groups, a bias would logically only make them learn better in one case or in the other (with a constant bias towards one of the colors). However, the experiment shows a lower CCR of both those groups compared to the spatial group. Hence, the bias resulting from prior experience is not likely to explain the low CCR. However, authors also demonstrate that the spatial and shock on yellow groups learn equally (see fig 2: Spatial X bin : Shock on yellow X bin) while shock on blue group do not learn. This might be due to a bias resulting from more experience feeding on blue than yellow flowers, hence difficulties to associate the blue color to an aversive stimulus.

- Line 523, the statement about sensitization and habituation is not clear.

- Line 526, are paper wasps known antagonists of honey bees? If not, there might not be any reason to expect an antagonistic behavior with paper wasps.

- Line 526, syntax.

- Line 540, the authors describe an alternative punishment method without much precision. This could be elaborated further for better understanding.

- Line 565, “collecting” should be replaced by “collected”.

- Line 576, authors should replace “overall learning” for “preference level”. The learning performances have been demonstrated to be similar.

- Line 587, as the intended negative controls (dead animal stimuli, compared to live ones) did show the same reduction of overall CCR compared to spatial group, this study would have benefitted from the use of additional controls to isolate the parameter that induces the reduction of CCR. For example, some studies use dummies to control for the possible chemical detection of the stimulus animal. In absence of such control (demonstrating no effect on behavior compared to the spatial group), one could argue that the measured effect (decrease in CCR levels) could be due to unintended experimental bias. This could be discussed somehow.

- Line 591, the earlier version of the visual APIS setup (Kirkerud et al., 2017) already used multiple IR beams to assess position and velocity of bees in the apparatus.

- Line 592, Marchal and colleagues, 2019 also used a video analysis method for precise honey bee tracking inside a visual aversive shuttle box.

- Line 592-594, the idea developed by the authors in this sentence would benefit from a more explicit statement in order to be understandable without having to read the cited paper.

6. PLOS authors have the option to publish the peer review history of their article (what does this mean?). If published, this will include your full peer review and any attached files.

Reviewer #1: No

Reviewer #2: Yes: Paul Marchal

---

## [Author Response · Author response to Decision Letter 0]

11 Dec 2019

We are grateful for the insightful comments of the reviewers (see our acknowledgements). We have substantially revised our manuscript based on the comments of the editor and reviewers. We address specific comments in our response to reviewers document.

---

## [Decision Letter · Decision Letter 1]

30 Dec 2019

PONE-D-19-27434R1

Conspecific and interspecific stimuli reduce initial performance in an aversive learning task in honey bees (Apis mellifera)

PLOS ONE

Dear Dr. Varnon,

Thank you for submitting your manuscript to PLOS ONE. After careful consideration, we feel that it has merit but does not fully meet PLOS ONE’s publication criteria as it currently stands. Therefore, we invite you to submit a revised version of the manuscript that addresses the points raised during the review process.

Please address the statistical concerns of Reviewer 1.

We would appreciate receiving your revised manuscript by Feb 13 2020 11:59PM. To enhance the reproducibility of your results, we recommend that if applicable you deposit your laboratory protocols in protocols.io, where a protocol can be assigned its own identifier (DOI) such that it can be cited independently in the future. For instructions see: http://journals.plos.org/plosone/s/submission-guidelines#loc-laboratory-protocols

We look forward to receiving your revised manuscript.

Kind regards,

James C. Nieh, Ph.D.

Academic Editor

PLOS ONE

Reviewers' comments:

Reviewer's Responses to Questions

**Comments to the Author**

1. If the authors have adequately addressed your comments raised in a previous round of review and you feel that this manuscript is now acceptable for publication, you may indicate that here to bypass the “Comments to the Author” section, enter your conflict of interest statement in the “Confidential to Editor” section, and submit your "Accept" recommendation.

Reviewer #1: (No Response)

Reviewer #2: All comments have been addressed

2. Is the manuscript technically sound, and do the data support the conclusions?

Reviewer #1: Partly

Reviewer #2: Yes

3. Has the statistical analysis been performed appropriately and rigorously? 

Reviewer #1: No

Reviewer #2: Yes

4. Have the authors made all data underlying the findings in their manuscript fully available?

Reviewer #1: Yes

Reviewer #2: Yes

5. Is the manuscript presented in an intelligible fashion and written in standard English?

Reviewer #1: Yes

Reviewer #2: Yes

6. Review Comments to the Author

Reviewer #1: Although I appreciate the re-analysis of the data, and the fact that the authors have responded to several of my comments, I do not feel that some of the more important ones have been addressed sufficiently. For example, it should not be up to the reader to decide if and how to correct for multiple comparisons. However, this is a relatively minor quibble. My more major concern is that you are arguing for a particular interpretation of the data, and you should therefore back it up. The graphs alone, especially lacking any sort of error bar, do not tell a compelling enough story. Outside the difference between the spatial group and all other groups, and the general learning curve, I cannot see patterns clear enough in these data to warrant drawing definitive conclusions or dispensing with p-values, arbitrary as they may be. And the statistics you have provided do not strongly support your interpretation, either, in my opinion.

In my previous review, I said: "They claim that, although the stimulus organism was noticed, the presence of another organism seems to have decreased performance, especially at the beginning of a trial, when subjects often performed below chance level. This is puzzling, given that what constituted "correct" was different across conditions -- so would have to mean that sometimes, the subject got closer to the stimulus animal, sometimes, further away (sometimes one side of the box, sometimes the opposite side), but that they seem to have been drawn toward making an incorrect choice in all cases." I was unable to find a place in the text or the response that dealt with this issue.

Ultimately, as I said in my first review, I think this was a well-conducted (and no doubt time-consuming) study; however, I do not believe the data, as they are presented in their current form, are strong enough to justify publication.

Reviewer #2: The revised version of the manuscript is, in my opinion, totally satisfactory. The authors addressed all major and minor comments and suggestions from both reviewers in a comprehensive and positive manner that led to the improvement of the manuscript.

7. PLOS authors have the option to publish the peer review history of their article (what does this mean?). If published, this will include your full peer review and any attached files.

Reviewer #1: No

Reviewer #2: Yes: Paul Marchal

---

## [Author Response · Author response to Decision Letter 1]

7 Jan 2020

We appreciate the efforts of the reviewers, especially in regard to speed. We have revised the manuscript and we respond to specific comments below in the response to reviewers document.

---

## [Editor Report · Decision Letter 2]

9 Jan 2020

Conspecific and interspecific stimuli reduce initial performance in an aversive learning task in honey bees (Apis mellifera)

PONE-D-19-27434R2

Dear Dr. Varnon,

We are pleased to inform you that your manuscript has been judged scientifically suitable for publication and will be formally accepted for publication once it complies with all outstanding technical requirements.

With kind regards,

James C. Nieh, Ph.D.

Academic Editor

PLOS ONE
---

## [Editor Report · Acceptance letter]

15 Jan 2020

PONE-D-19-27434R2 

Conspecific and interspecific stimuli reduce initial performance in an aversive learning task in honey bees (*Apis mellifera*) 

Dear Dr. Varnon:

I am pleased to inform you that your manuscript has been deemed suitable for publication in PLOS ONE. Congratulations! Your manuscript is now with our production department. 

With kind regards,

on behalf of

Dr. James C. Nieh 

Academic Editor

PLOS ONE